ð | Open Peer Review | Computational Biology | Research Article
# Genome-centric analyses of 165 metagenomes show that mobile genetic elements are crucial for the transmission of antimicrobial resistance genes to pathogens in activated sludge and wastewater

Nafi'u Abdulkadir,[1,2] Joao Pedro Saraiva,[1] Junya Zhang,[3,4] Stefan Stolte,[5] Osnat Gillor,[6] Hauke Harms,[1,2] Ulisses Rocha[1]

**ABSTRACT** Wastewater is considered a reservoir of antimicrobial resistance genes (ARGs), where the abundant antimicrobial-resistant bacteria and mobile genetic elements facilitate horizontal gene transfer. However, the prevalence and extent of these phenomena in different taxonomic groups that inhabit wastewater are still not fully understood. Here, we determined the presence of ARGs in metagenome-assembled genomes (MAGs) and evaluated the risks of MAG-carrying ARGs in potential human pathogens. The potential of these ARGs to be transmitted horizontally or vertically was also determined. A total of 5,916 MAGs (completeness >50%, contamination <10%) were recovered, covering 68 phyla and 279 genera. MAGs were dereplicated into 1,204 genome operational taxonomic units (gOTUs) as a proxy for species ( average nucleotide identity >0.95). The dominant ARG classes detected were bacitracin, multi-drug, macrolide-lincosamide-streptogramin (MLS), glycopeptide, and aminoglycoside, and 10.26% of them were located on plasmids. The main hosts of ARGs belonged to *Escherichia*, *Klebsiella*, *Acinetobacter*, *Gresbergeria*, *Mycobacterium*, and *Thauera*. Our data showed that 253 MAGs carried virulence factor genes (VFGs) divided into 44 gOTUs, of which 45 MAGs were carriers of ARGs, indicating that potential human pathogens carried ARGs. Alarmingly, the MAG assigned as *Escherichia coli* contained 159 VFGs, of which 95 were located on chromosomes and 10 on plasmids. In addition to shedding light on the prevalence of ARGs in individual genomes recovered from activated sludge and wastewater, our study demonstrates a workflow that can identify antimicrobial-resistant pathogens in complex microbial communities.

**IMPORTANCE** Antimicrobial resistance (AMR) threatens the health of humans, animals, and natural ecosystems. In our study, an analysis of 165 metagenomes from wastewater revealed antibiotic-targeted alteration, efflux, and inactivation as the most prevalent AMR mechanisms. We identified several genera correlated with multiple ARGs, including *Klebsiella*, *Escherichia*, *Acinetobacter*, *Nitrospira*, *Ottowia*, *Pseudomonas*, and *Thauera*, which could have significant implications for AMR transmission. The abundance of *bacA*, *mexL*, and *aph(3")-I* in the genomes calls for their urgent management in wastewater. Our approach could be applied to different ecosystems to assess the risk of potential pathogens containing ARGs. Our findings highlight the importance of managing AMR in wastewater and can help design measures to reduce the transmission and evolution of AMR in these systems.

**KEYWORDS** antimicrobial resistance genes, metagenome-assembled genomes, mobile genetic elements, wastewater, activated sludge, pathogens, resistome

Address correspondence to Ulisses Rocha, ulisses.rocha@ufz.de.

The authors declare no conflict of interest.

See the funding table on p. 19.

Antimicrobial resistance (AMR) is a serious global health issue that requires urgent attention (1) as it significantly threatens public health and the global economy (2). By 2030, the global impact of AMR on the economy is expected to exceed $2 trillion due to the higher costs of second-line drugs and treatment failures (1, 2). In addition to its impact on medical communities, AMR has significant implications for public health and the environment due to its occurrence in soils, wastewater (WW) and sewage. In 2019, 4.95 million deaths were linked to bacterial AMR (3), and this number is projected to reach 10 million per year by 2050, provided that no appropriate measures to reduce the transmission of AMR genes are implemented (3–6). The emerging problem of AMR and antimicrobial-resistant bacteria (ARBs) is not only limited to human health but also concerns animals and ecosystems, making AMR a vital component of environmental pollution to be addressed (1). Thus, it is crucial to understand the distribution and potential hosts of antimicrobial resistance genes (ARGs) in different environments, including wastewater treatment plants (WWTPs) and activated sludge (AS), to prevent further spread and /or development of new AMR mechanisms in microbes (7).

Municipal WWTPs receive large amounts of pharmaceutical pollutants (including antibiotics) and pathogens (7–10). Many WWTPs rely on AS, a densely rich and diverse microbial community, to remove organic matter. The AS can also play a role in the biodegradation of organic and inorganic pollutants (10). However, current WWTPs are limited in their ability to handle antibiotics, ARBs, and ARGs, which may persist in the effluent (7, 11). WWTPs are considered an environmental hotspot for disseminating ARGs (9, 11, 12) due to the presence of mobile genetic elements (MGEs) and diverse ARBs. The high abundances of bacterial communities in the WW facilitate the direct route for the distribution of ARGs in the environment (13, 14). Yet, the link between the ARGs detected in the WW flow and their hosts was rarely established. Therefore, understanding the prevalence of ARGs in pathogenic and non-pathogenic bacteria is paramount for identifying factors driving the dissemination of ARGs from WWTPs to the environments that receive the effluent. Previous studies have shown that WWTPs are very important sources of resistance genes due to the secretion of antibiotic residues from human waste, veterinary sources, and hospital (9, 13, 15). Therefore, surveillance of ARGs is necessary in WWTP as part of the effort to diminish the emergence and distribution of resistance in the ecological environment and the possibility of detecting new bacterial resistance mechanisms. It is also essential to understand the various mechanisms by which bacterial species develop resistance to antibiotics for the establishment of policies to fight resistance.

Additionally, previous studies have reported the occurrence of ARGs in the treated WW and their distribution into the receiving environment, which further reveals that continuous discharge of poorly treated WW can enable the transfer of resistance genes to pathogenic bacteria and spread ARBs in the environments (15, 16). The existence of ARGs and ARBs in the activated sludge, effluents, and influents of WWTPs from various countries shows the global distribution of resistance genes in the environment (13, 15–17). A recent review by Nava and co-workers revealed that multi-resistance bacteria are present in WW and distributed in the environment through effluent discharge, which may lead to the development of "superbug" species (13). The existence of multi-resistance species in WW is supported by the co-association of antibiotics, heavy metals, ARBs, and ARGs in WWTPs (16). Therefore, an urgent need to monitor heavy metal resistance in the WWTP and to design adequate strategies for assessing the risks of ARGs and heavy metal resistance in ecological settings is necessary.

One of the challenges of AMR research is the lack of a standard method for quantification and surveillance of ARG acquisition, despite the richness of ARGs in the WW effluents and transmission of genes from pathogens to commensal species in the environment facilitated by MGEs via horizontal gene transfer (HGT) or vertical gene transfer (VGT) (13). The studies of Nava et al. (13) and Larsson and Flach (14) further revealed no standard method for removing ARG in the environment, including WWTPs. Therefore, strategies involving biotic and microbial remediations are needed to mitigate

the evolutionary selection of ARGs. In an effort to tackle the menace of ARGs in the environment, public awareness about the reasonable usage and pernicious upshot of antibiotic misuse and abuse should be implemented to reduce antibiotic dissipation.

The development of high-throughput sequencing technologies has made it possible to sequence the entire DNA content of microbial communities, a process known as metagenomics (18–20). Metagenomics has drastically expanded our knowledge of uncultivable microorganisms and their functional potential (18, 21, 22). Additionally, this approach improved the taxonomic classification of microbial communities up to the species level while allowing the discovery of novel functional genes in the microbiome (22, 23). In recent years, metagenomics and bioinformatics have become approaches powerful in identifying ARGs and understanding AMR mechanisms of microbial communities in aquatic and terrestrial environments (11, 23–26). Furthermore, metagenomics can determine whether ARGs are located in chromosomes or on MGEs such as plasmids (4), integrases (27), and transposons (28), which are responsible for disseminating ARGs in the environment through HGT. Studies on metagenomics analysis unveiled the presence of important clinical ARG classes in activated sludge, including penicillin, tetracycline, sulfonamides, and others that remain in treated wastewater (25, 29). A study by Cacace et al. (30) showed the abundance of ARGs in treated WW and receiving bodies from WWTP effluent in 10 European countries. The study showed the presence of ARGs in all effluents and river water samples, demonstrating a complex method of acquiring ARGs in different bacterial communities. Recently, Talat and co-workers (31) provided a comprehensive overview of ARGs in hospital wastewater using metagenomics. The study uncovered many important clinical resistance gene classes, including beta-lactam, aminoglycosides, macrolide carbapenem, and sulfonamides, which were hosted by human pathogens, such as *Pseudomonas aeruginosa*, *Acinetobacter baumannii*, and *Klebsiella pneumoniae*, evinced the peril associated with ARG transmission in the environments (31).

Recovery of metagenome-assembled genomes (MAGs) makes it possible to study the distribution and evolution of ARGs and gain insight into the complex interaction between ARGs and their microbial hosts (32, 33). Identifying hosts of MGEs is essential to understanding the transmission of resistance genes within microbial communities (34). MAGs may help identify MGEs in hosts, such as plasmids. Identification of plasmids in a given host can be challenging because plasmids are self-replicating (35) and can be hosted by more than one species (34, 36). Notably, plasmids are known to be significant vectors for gene transfer in many bacterial populations than any other mobile genetic elements (34, 37). Although previous studies suggested that genetic transfer of ARGs occurred between closely related species (38), recent findings have shown that this phenomenon occurs between distant phylogenetic taxa (39).

Additionally, MAGs could permit the prediction of virulence factor genes (VFGs), giving insight into the relationship between ARG carriers and pathogens. VFGs are essential molecules that determine the pathogenicity of microbiomes and their ability to cause disease in hosts such as humans (40–42). Recently, genome-centric approaches have been used to determine the distribution patterns of resistance genes in urban, fermented foods and sewage to manage ARGs as a public health problem. A previous study by Tan and colleagues (43) identified the distribution patterns of ARGs in sufu bacterial communities. Furthermore, it revealed ARGs were enriched in opportunistic pathogens. The study revealed the presence of the most critical clinical drug classes used to treat diseases. It showed that MAGs uncovered significant pathways of human resistance gene consumption due to the uptake of ready-to-eat food (43). The study by Zhang et al. revealed the profile of ARGs in combined sewage overflows and recovered MAGs conferring resistance to many ARGs, including *bacA*, *acrA*, *rsmA*, and *mexK OmpA* (4).

In this study, our objectives were to (i) assess the presence and abundance of ARGs in WW and AS, (ii) determine the distribution patterns of ARGs and MGEs in microbial communities, (iii) link the VFGs and ARGs in binned MAGs, and (iv) assess the risk of

ARGs based on the co-occurrence of VFGs in the same MAGs. We hypothesized that the co-localization of ARGs on mobile genetic elements such as plasmids in microbial genomes could increase the acquisition of virulence factor genes, resulting in potential HGT to commensal organisms. To test our hypothesis, we used publicly available metagenomics data to recover MAGs for genome-centric analysis. We compared ARGs and plasmids within different taxonomic groups and examined the risk of transfer of ARGs to potential human pathogens. We identified ARGs and VFGs in MAGs, determined their carriers (plasmids or chromosomes), and identified their hosts. Our results showed that ARGs were prevalent in bacteria identified as human pathogens. One identified species is classified as a critical pathogen by the World Health Organization (WHO). However, we could not generate beta-diversity inferences due to disparities in the number of samples and sequencing depth; instead, we concentrated on comparing the resistome in various taxonomic groups.

## RESULTS

### Recovery of metagenome-assembled genomes in activated sludge and wastewater samples

We selected 138 AS metagenomes and 27 WW treatment plants metagenomes. These metagenomes were collected from 13 locations across Europe, Asia, Australia, and North and South America (Fig. 1). Most of the samples were recovered from municipal WW treatment plants (55 in total), AS from WW treatment plants (37 in total), AS from domestic WW treatment plants (22 in total), and full-scale industrial WW treatment plants (12 in total) (Table S1). The numbers of samples collected from each geographic location are shown in Table 1. Only 28 of the selected metagenomes had been published in research articles (Table S1). We retrieved $4.8 \times 10^9$ sequence reads in the metagenomes selected from the Sequence Read Archive (SRA) database. These sequences were distributed in 165 libraries with an average of $2.88 \times 10^7$ ($1.82 \times 10^6$ to $8.62 \times 10^7$ reads per library). The number of base pairs in the metagenomes was $1.37 \times 10^{12}$ with an average of $8.22 \times 10^9$ ($6.9 \times 10^8$ bp to $2.6 \times 10^{10}$ bp). Furthermore, we also retrieved the library fragment length of metagenomes that ranged from 150 to 602 bp with an average of 284.08 (Table S2). The total numbers of reads, base pairs, and average fragment length of metagenomes from AS and WW are shown in Table S2. Of the 5,916 recovered MAGs, almost 95% were derived from AS samples (Table S2). A quality assessment analysis of the MAGs showed completeness and contamination of 50.08%–100.0% and 0%–8.79%, respectively, with quality scores ranging from 50.03 to 100.00.

We identified a total of 2,660 (44.96%) high-quality MAGs (completeness >90% and contamination <5%), while we classified 3,256 (55.04%) as medium-quality MAGs (completeness ranging from 50% to 89% and contamination <10%) (Fig. 2A). Further details on completeness, contamination, N50 statistics, strain heterogeneity, genome size, and the number of contigs of MAGs are provided in Table S3. We dereplicated our 5,916 MAGs into 1,204 genome operational taxonomic units (gOTUs) using average nucleotide identity distances greater than 0.95 as a proxy of species. The genomic features of the gOTUs, including completeness, contamination, and quality score of the gOTUs, are provided in Fig. S1 and Table S4.

### Taxonomy and abundance of MAGs recovered from AS and WW treatment plants

The taxonomy to the 5,916 MAGs was assigned using GTDB-Tk (44). Our data set showed 5,668 bacteria and 248 archaea encompassing 61 and 8 phyla, respectively (Fig. 2B; Table S5). The 10 most abundant bacterial phyla comprised 70.58% of all recovered MAGs (Table S6). The results also showed that 5–15 families represented these abundant phyla, except Proteobacteria and Bacteroidota, represented by 52 and 22 families, respectively (Table S6). Our analyses showed 67 and 22 phyla were found in AS and WW, respectively. Furthermore, 21 phyla were shared between MAGs recovered from AS and WW samples (Fig. S2).

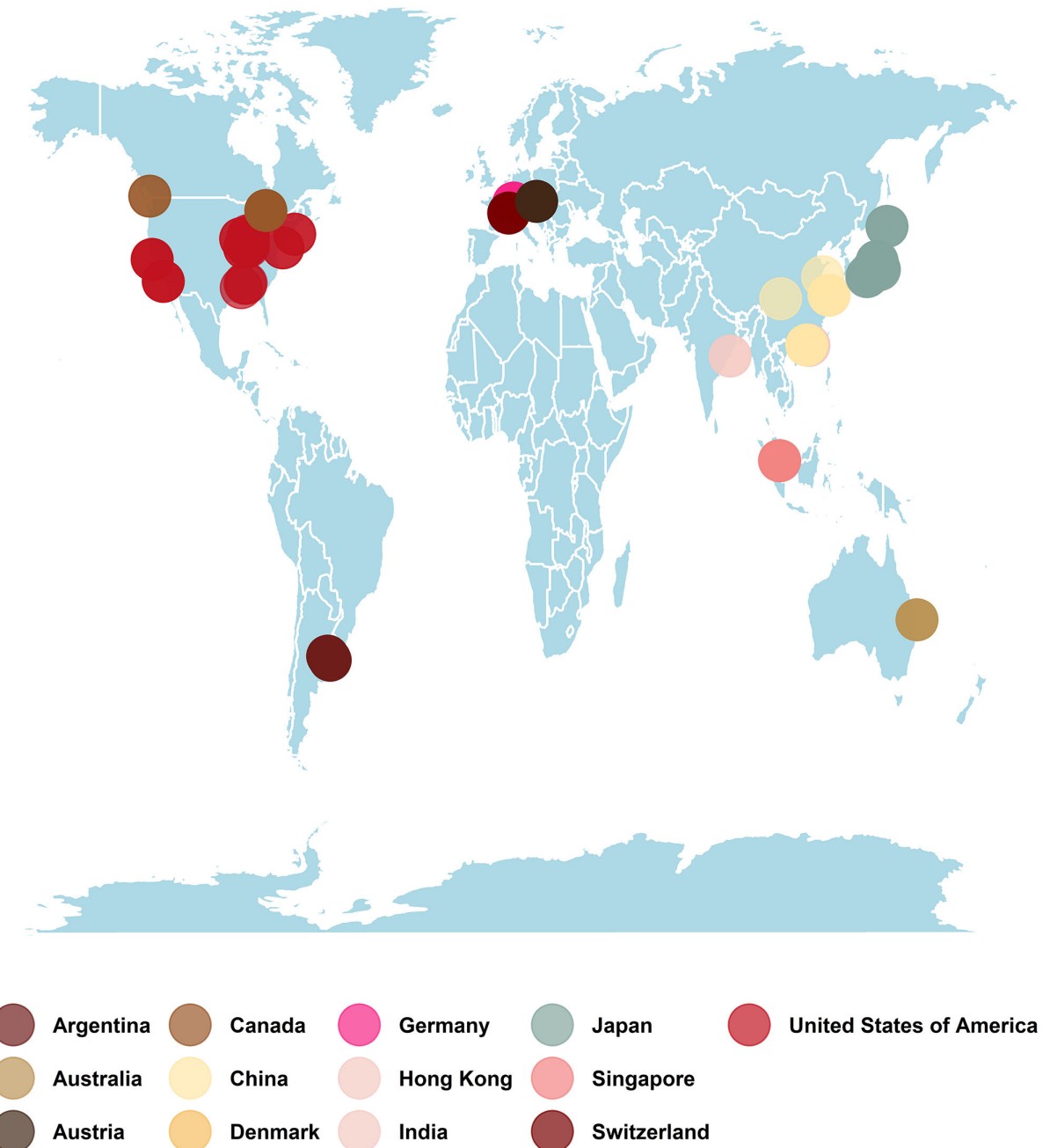

**FIG 1** Map showing the geographical distribution of samples collected from different locations. A total of 165 samples were collected, covering 13 different locations and 5 continents. The map was generated with Tidyverse package in R using geographical coordinates (longitude and latitude) of sample metadata retrieved from SRA.

At the class level, the most dominant taxa were Bacteroidia (1,224), Gammaproteobacteria (662), Clostridia (257), Anaerolineae (237), and Syntrophia (179). Further classification showed that eight orders and seven families of bacteria dominated the communities (Tables S5 and S6). More than 70.69% of the MAGs were not classified at the genus level (Table 2), while only 5.61% of the MAGs were assigned to a species (Table 2; Table S6).

Regarding archaea, the results showed that Halobacteriota (158) was the most abundant phylum. Within the archaeal MAGs, 17 known genera and 7 known species were identified (Fig. 2C). We recovered 247 archaeal MAGs from AS, of which 8 different

**TABLE 1** Number of metagenomes collected from different countries and sample sources (activated sludge or wastewater)

| Country | Sample source | | Total |
| --- | --- | --- | --- |
| | **Activated sludge** | **Wastewater** | |
| Argentina | 14 | 0 | 14 |
| Austria | 34 | 0 | 34 |
| Australia | 2 | 2 | 4 |
| Canada | 2 | 7 | 9 |
| China | 8 | 6 | 14 |
| Denmark | 1 | 0 | 1 |
| Germany | 0 | 8 | 8 |
| Hong Kong | 11 | 2 | 13 |
| India | 0 | 2 | 2 |
| Japan | 17 | 0 | 17 |
| Singapore | 20 | 0 | 20 |
| Switzerland | 4 | 0 | 4 |
| USA | 25 | 0 | 25 |
| Total | 138 | 27 | 165 |

phyla were identified. In contrast, only a single archaeal MAG was recovered from WW, classified as Microarhaeota. Additionally, Microarhaeota was the only archaeal phylum found in both AS and WW.

## Deciphering ARGs in activated sludge and wastewater

We detected the presence of ARGs in the samples collected from AS and WW. The AMR genes of AS and WW varied with respect to the number and types of AMR classes in each sample. Nine AMR classes and 22 ARGs represented the AS resistome. The AMR classes of dimethylpyrimidine, glycopeptide, sulfonamide, and tetracycline were found exclusively in the AS genomes (Table S7). On the other hand, we found 6 AMR classes and 19 ARGs in the WW samples. Fosfomycin was detected exclusively in WW samples (Fig. S3B). The most abundant ARG classes in WW conferred resistance to multiple drugs (37.93%) and bacitracin (34.48%). In contrast, in AS, the most abundant classes conferred resistance to bacitracin (29.16%), multiple drugs (18.75%), and glycopeptide and MLS (16.67%), as shown in Fig. S4B. The difference in AMR classes between AS and WW was not statistically significant ($P = 0.607$, as revealed by independent $t$-test). Five classes were shared between AS and WW, including resistance to aminoglycoside, bacitracin, beta-lactam, MLS, and multiple drugs (Fig. S3B).

Twenty-four AMR families were identified in our data set. The most prevalent gene families in the WW and AS MAGs were undecaprenyl pyrophosphate-related proteins, resistance-nodulation-cell division antibiotic efflux pump APH (3″), and msr-type ABC-F protein (Table S8), while the glycopeptide resistance gene cluster was found only in AS MAGs (Table S8). The most common resistance mechanisms in AS and WW were antibiotic efflux pumps and alteration of the antibiotic target (Table S8). Other AMR mechanisms were also detected, including target replacement, antibiotic inactivation, antibiotic target protection, antibiotic target replacement, and reduced antibiotic permeability.

## Geographical distribution of the ARGs

Metagenomes were retrieved from 13 locations on 5 continents (Fig. 1). A total of 10 AMR classes and 1 unclassified resistance gene class were found at different locations. Only one AMR class was found in samples from Argentina, while samples from the remaining locations contained multiple ARGs. Furthermore, MAGs recovered from samples collected from different locations were found to encode multi-drug resistance gene classes, except in Japan and the USA. The diversity and abundance of resistance gene

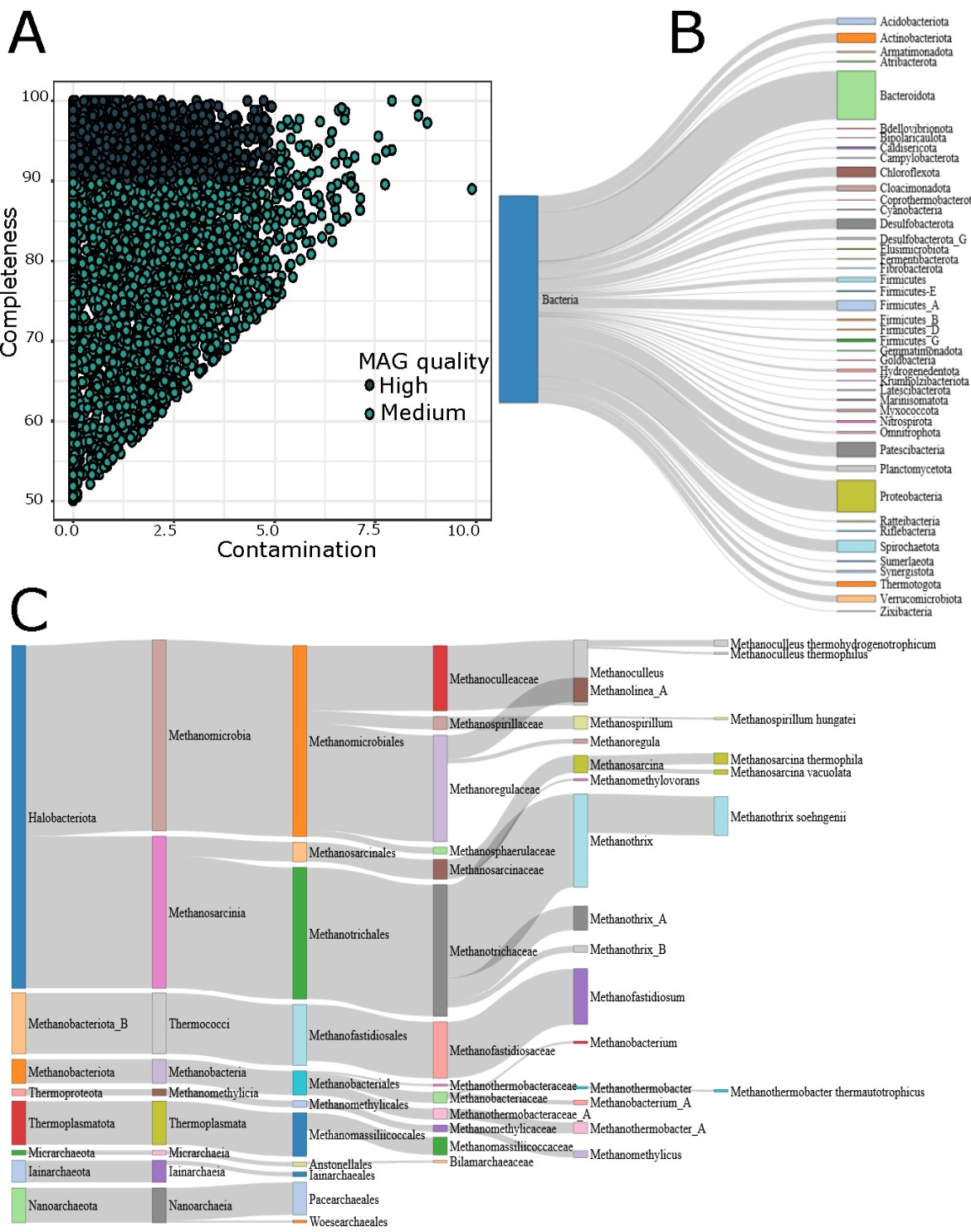

**FIG 2** Metagenome-assembled genome (MAG) quality and taxonomy. (A) A scatter plot showing the contamination and completeness levels of the 5,916 MAGs. Each point is colored according to its quality score. The quality score is calculated as the MAG completeness minus five times its contamination [% completeness − (5 × % contamination)]. Medium-quality MAGs have completeness higher than 50% and contamination lower than 10%. High-quality MAGs have completeness higher than 90% and contamination lower than 5%. All MAGs have a quality score higher than or equal to 50. (B) Sankey plot showing the taxonomic diversity of the bacterial communities recovered from the AS and WW defined by GTDB-Tk. All phyla containing less than five MAGs were removed from the figure. (C) The Sankey plot shows the archaea community's taxonomic diversity at different taxonomic levels recovered from the AS and WW.

classes varied between locations (Table S7): multi-drug (seven locations) and bacitracin (six locations) resistance gene classes were widespread.

**TABLE 2** Number of MAGs[b] with known and unknown phylogenetic classifications (defined by GTDB-Tk[a]) at different taxonomic levels[c]

|  | Total | MAGs with known classification n (%) | MAGs with unknown classification n (%) |
|---|---|---|---|
| Domain | 2 | 5,916 (100.00) | 0 (0.00) |
| Phylum | 68 | 5,801 (99.06) | 115 (1.94) |
| Class | 96 | 5,252 (88.77) | 664 (11.23) |
| Order | 155 | 4,436 (74.98) | 1,480 (25.02) |
| Family | 211 | 3,076 (51.99) | 2,840 (48.01) |
| Genus | 279 | 1,734 (29.31) | 4,182 (70.69) |
| Species | 114 | 332 (5.61) | 5,584 (94.39) |

[a]Chaumeil P-A, Mussig AJ, Hugenholtz P, Parks DH. 2020. GTDB-Tk: a toolkit to classify genomes with the Genome Taxonomy Database. Bioinformatics 36:1925–1927.
[b]MAG, metagenome-assembled genome.
[c]The number between parenthesis represents the percentage of MAGs for the different taxonomic levels with known and unknown taxonomic classifications.

In terms of the number of multiple resistance gene classes found in a single country, the USA had the highest number and abundance of ARGs, with 17 ARGs and 5 AMR classes, followed by Germany with 15 ARGs and 6 AMR classes, and Singapore and China with 8 ARGs and 3 AMR classes (Table S7). Fosfomycin, diaminopyrimidine, sulfonamide, and tetracycline were rare in different locations, including China, Singapore, and the USA. The relative abundances of AMR classes and locations were visualized in a circle to calculate the hotspots of areas highly polluted with antibiotics (Fig. S3A).

## Prevalence and distribution of ARGs in microbial communities

We determined the distribution pattern of ARGs in our MAGs at different taxonomic levels. Our results showed that AMR classes could be divided into four groups, each corresponding to one of the four categories (ubiquitous, widespread, common, and sparse) described by Magnúsdóttir et al. (26). Genome-centric analysis revealed 10 unique AMR classes and 1 unclassified class containing 38 unique ARGs (Table S9). Our analysis indicated that Proteobacteria harbors six AMR classes (e.g., aminoglycoside, bacitracin, beta-lactam, fosfomycin, multi-drug, MLS, and the unclassified AMR class), while Actinobacteriota harbors only two AMR classes (glycopeptide and MLS). We also found that Bacteroidota, Firmicutes, and Firmicutes A were resistant to more than two antibiotic classes. At the same time, Firmicutes B was resistant only to glycopeptide, as shown in Fig. 3A.

The results revealed a high prevalence of ARGs in the Myxococcota, Verrucomicrobiota, and Proteobacteria phyla. We found *bacA*, *Mrx*, and *aph(3″)-I* at the highest prevalence in Myxococcota and Verrucomicrobiota. Proteobacteria hosted 24 different genes, including *bacA*, *emrD*, and *mexT*. Firmicutes (A and B) were host to eight resistance genes, including *aadE*, *bacA*, and *tet44*. The distribution pattern of ARGs across different taxa is provided in Fig. 3B and Table S9.

Additionally, we determined the frequency of MAGs that contained ARGs belonging to each AMR class. The results showed that bacitracin resistance was present at the largest number of MAGs and was presented with a single ARG in our data (Fig. S4A), while resistance to multi-drug and MLS was attributed to multiple ARGs. Kendall's tau correlation also indicated a strong relationship between the numbers of unique ARGs per AMR class (Kendall's tau 0.588). The number of MAGs containing each ARG within each AMR class can be found in Fig. S4A.

Our results suggest that specific genera of bacteria are more likely to harbor ARGs (Fig. 4). At the genus level, 35 genera belonging to 23 orders were found to host ARGs. The most abundant orders were Enterobacteriales, with 12 ARGs; Pseudomonadales,

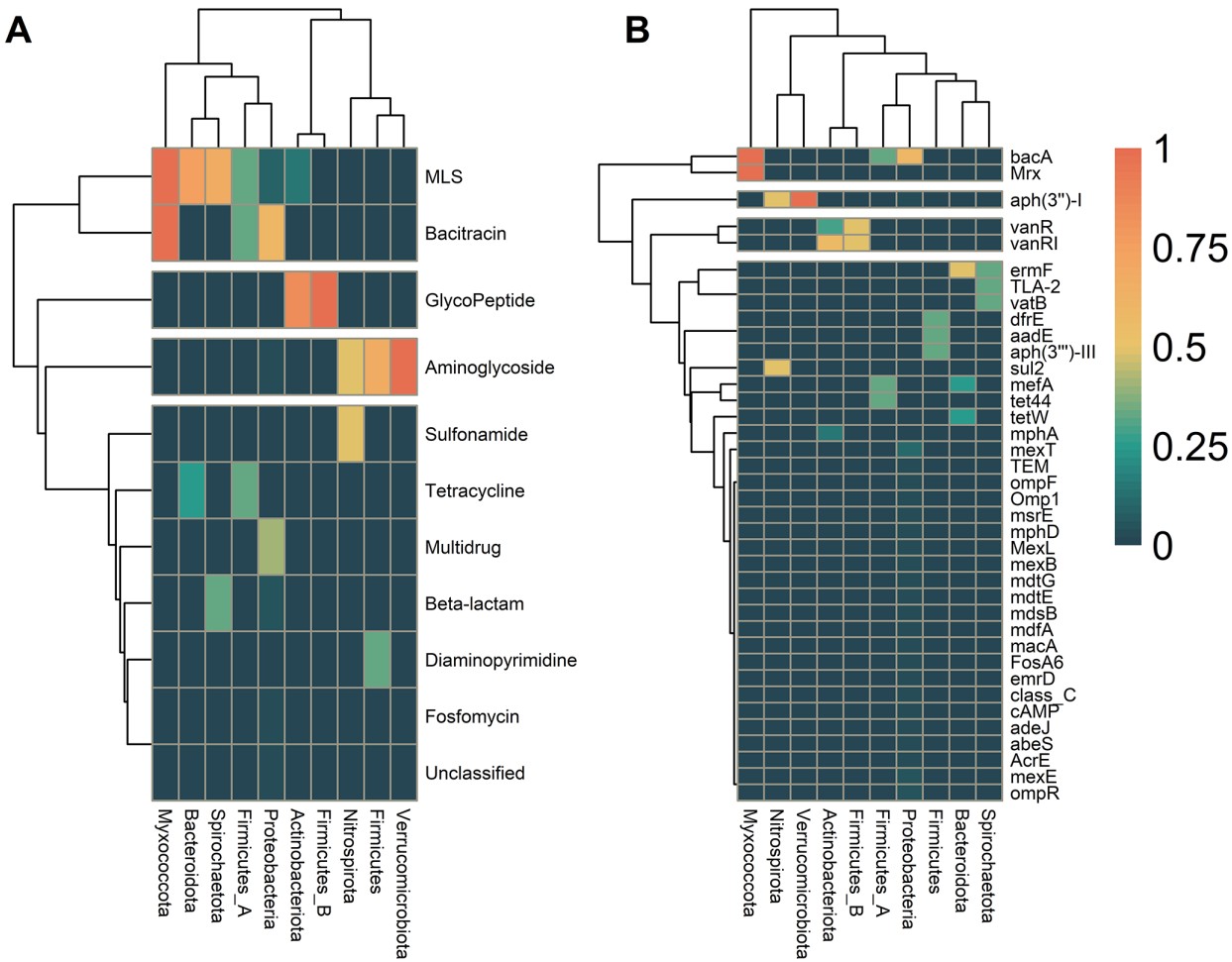

**FIG 3** Prevalence of antimicrobial resistance gene (ARG) classes in activated sludge (AS) and wastewater (WW) microbial communities. (A) Heatmap showing the prevalence of ARG classes in bacterial phyla recovered from AS and WW. (B) The prevalence of ARGs and the bacterial community at phylum level. The dendrogram is based on hierarchical clustering with Ward distance between the ARG class prevalence among the phyla.

with 8 ARGs; and Burkholderiales, with 5 ARGs (Fig S5). The three most abundant genera with the highest number of ARGs were *Acinetobacter*, *Escherichia*, and *Klebsiella*, each encoding resistance to multiple ARGs. The study also showed that some genera, such as *Giesbergeria*, host only one resistance gene (Fig. 4).

ARGs were hosted by different species. For instance, *bacA* was hosted by eight species, while *mexT* was hosted by two species (Fig. S6). The ARGs *tetW*, *tet44*, and *sul2* are among the clinical resistance genes according to the WHO (45); their hosts were only classified down to the genus level (Fig. S6).

## Mechanisms of ARGs

We used the Comprehensive Antibiotic Resistance Database (CARD) to determine the AMR mechanisms of specific ARGs. Our results showed that ARGs' most common resistance mechanisms were antibiotic target alteration, antibiotic efflux, and antibiotic inactivation (Table S8). Our analysis revealed that antibiotic efflux and inactivation were ubiquitous among ARGs (Fig. 5A). In contrast, resistance mechanisms such as antibiotic target alteration, protection, replacement, and membrane permeability reduction to antibiotics were rare among the detected ARGs (Fig. 5A). According to the CARD, the mechanisms of the remaining seven ARGs were undefined. The distributions of resistance mechanisms among bacterial genera were also analyzed. The results showed that antibiotic target alteration was ubiquitous among 13 genera; antibiotic efflux was

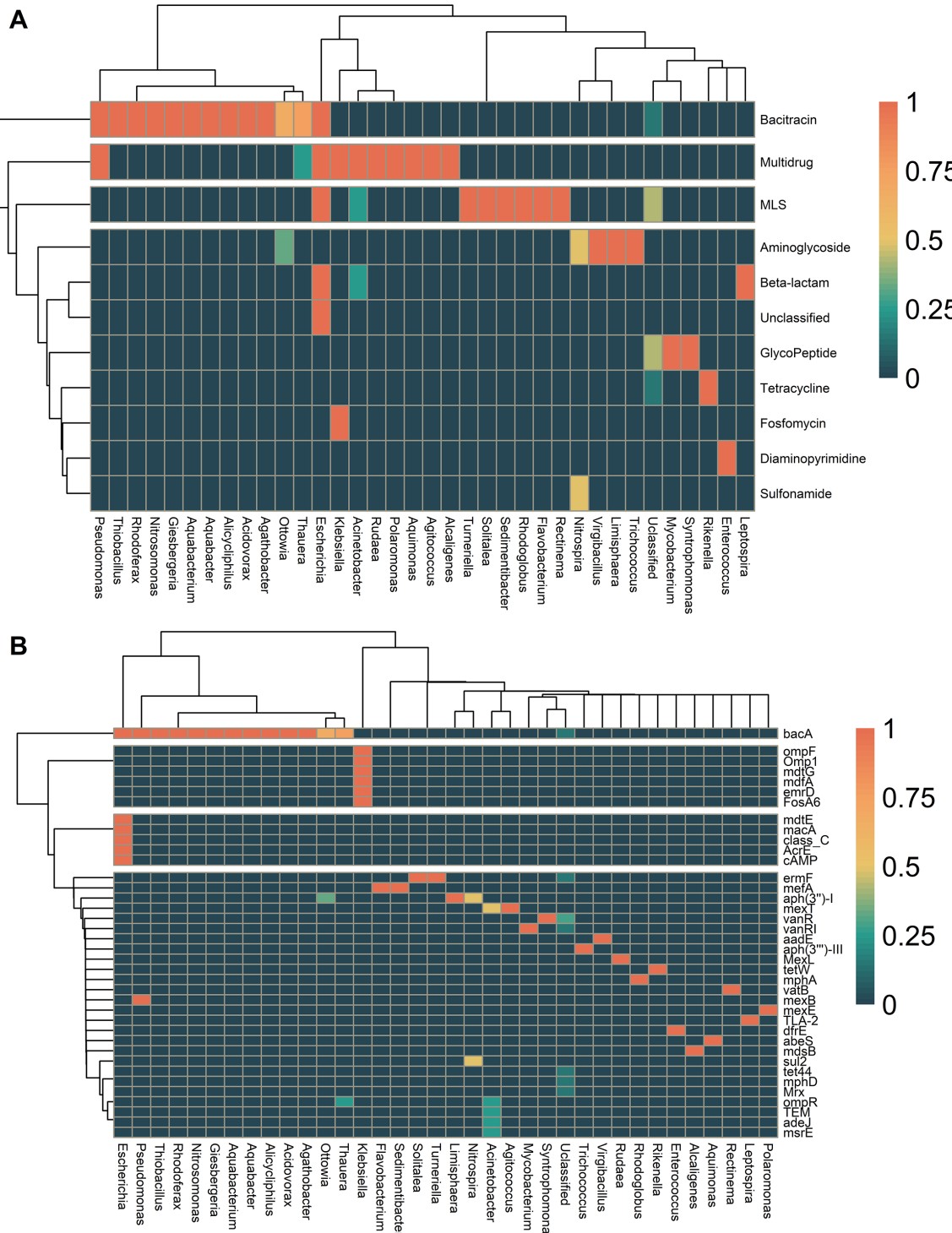

**FIG 4** Prevalence of antimicrobial resistance gene (ARG) classes in activated sludge (AS) and wastewater (WW) microbial communities. (A) Heatmap showing the prevalence of ARG classes in bacterial genera recovered from AS and WW. (B) The prevalence of ARGs and the bacterial community at genus levels. The dendrogram is based on hierarchical clustering with Ward distance between the ARG class prevalence among the genus.

widespread among seven genera; and the remaining genera had few occurrences of antibiotic target protection and undefined mechanisms (Fig. 5B).

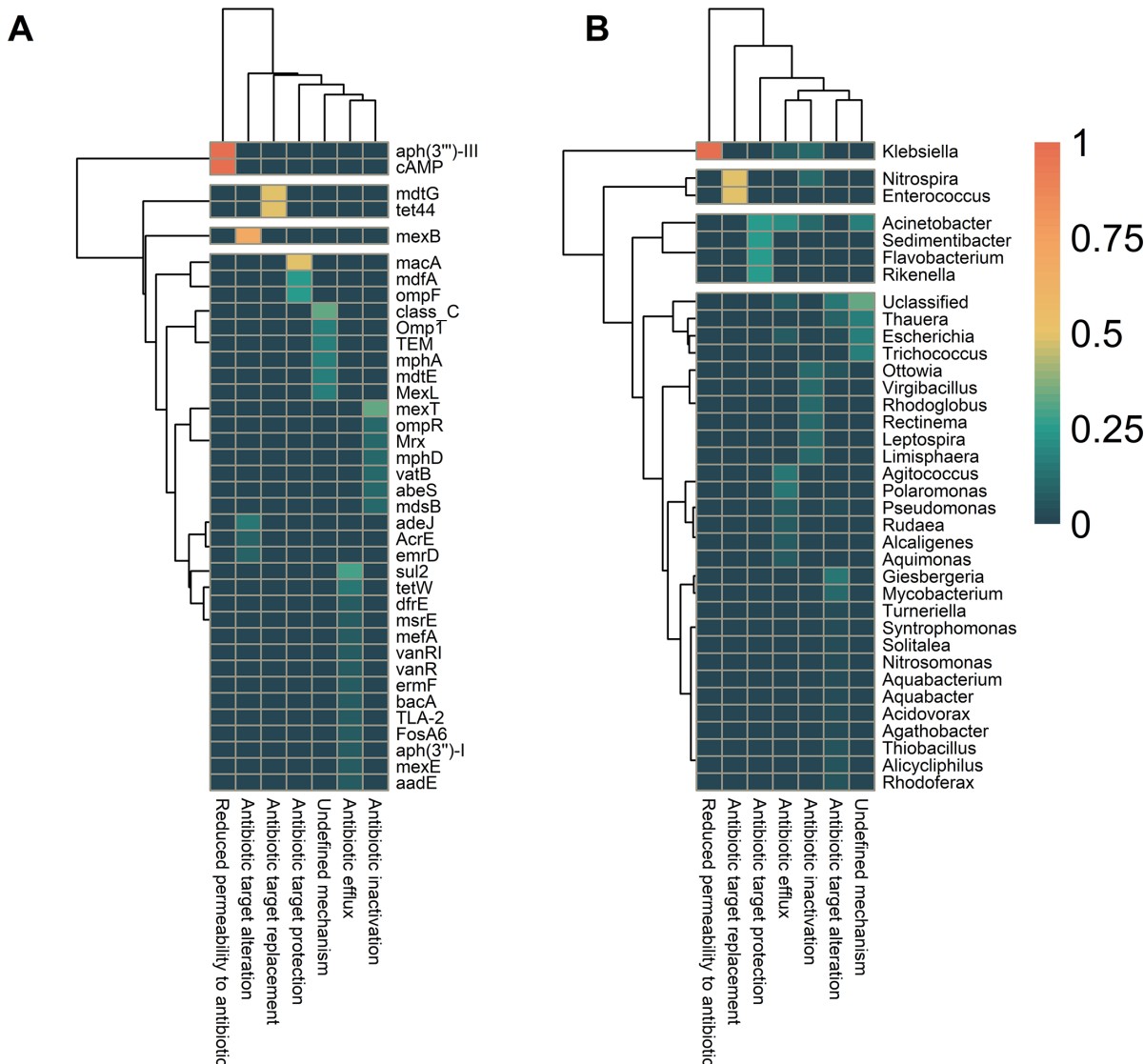

**FIG 5** Prevalence of resistance gene mechanisms within the different genera in our metagenome-assembled genome data set. (A) A heatmap showing the prevalence of different antimicrobial resistance gene (ARG) mechanisms found in each gene. The dendrogram is based on hierarchical clustering with Ward distance between the ARG resistance mechanism prevalence among the genes. (B) Resistance mechanisms. A heatmap showing the prevalence of antimicrobial resistance mechanisms within different genera recovered from the AS and WWTPs. The dendrogram is based on hierarchical clustering with Ward distance between the ARG class prevalence among the genera. cAMP, cAMP-regulatory_protein.

## Mobile ARGs in our MAG data set

This study determined the distribution of ARGs in chromosomes and MGEs, including plasmids. The dominant AMR classes in the chromosomes were bacitracin, followed by multi-drug, glycopeptide, and aminoglycoside. When comparing the number of AMR classes in plasmids and chromosomes (Fig. 6B), only MLS was found in both (Fig. 6A). The plasFlow model was used to predict whether a contig belongs to plasmids or chromosomes in the MAGs. A total of 57 ARGs were detected, of which 6 were assigned to plasmids and 51 to chromosomes (Table S10). The ARGs carried by the plasmids were clustered into three AMR classes: aminoglycoside, MLS, and sulfonamide.

We constructed a co-occurrence network of ARGs, MGEs, and bacterial genera to visualize the potential HGT in bacterial communities (Fig. 6C). The analysis showed significant positive correlations between bacterial genera and ARGs against eight antibiotics and one unclassified class. The co-occurrence results showed that *Klebsiella*

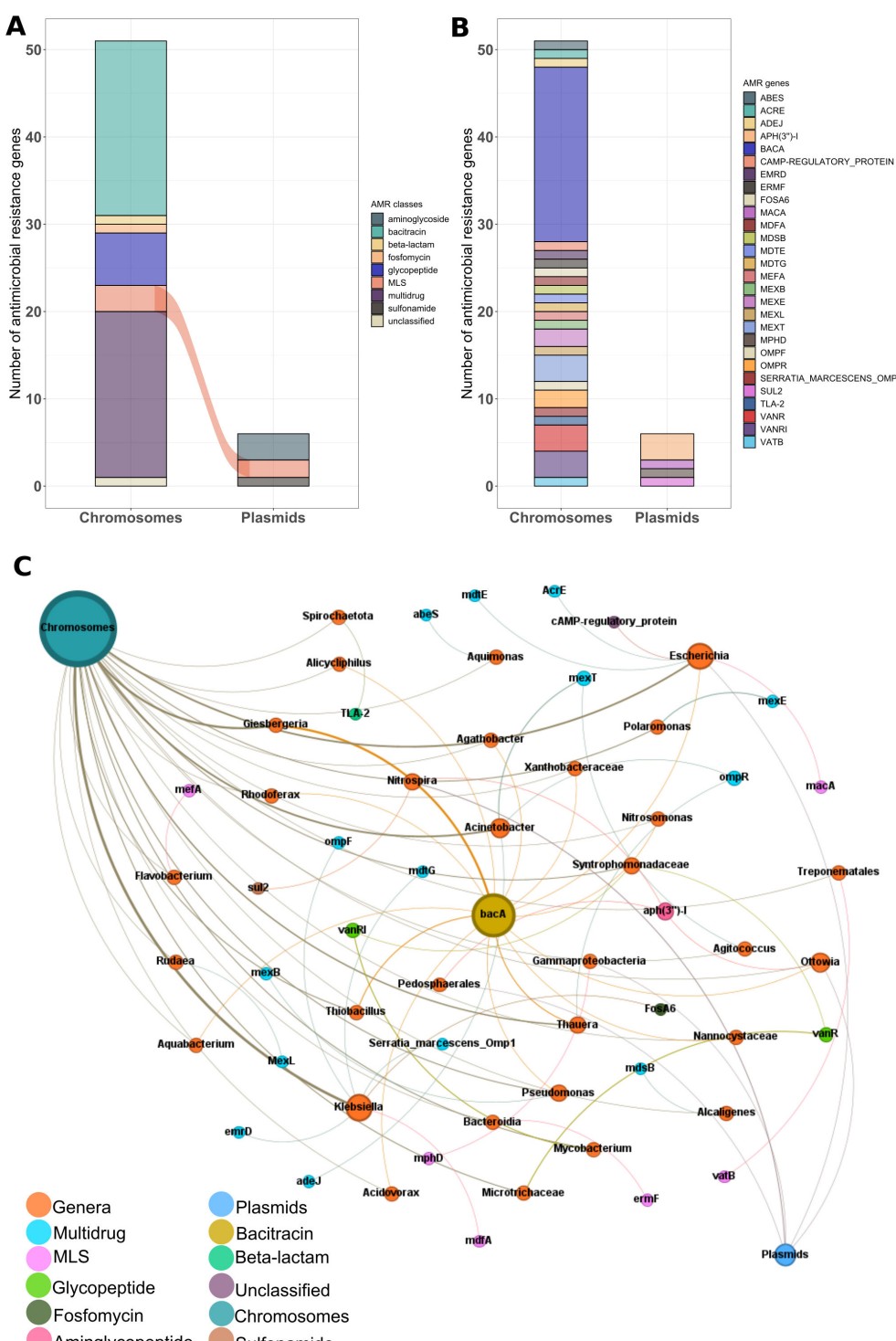

**FIG 6** The abundance of antibiotic resistance genes and co-occurrence network. (A) Bar plot showing the ARG classes' frequency encoded by plasmids and chromosomes. (B) Bar plot showing the frequency of ARGs located inside plasmids and chromosomes. (C) Network showing the host range (genus) and co-occurrence between ARGs found in chromosomes and mobile genetic elements such as plasmids.

correlated with six ARGs. *Escherichia* contributed the most to the ARG pool by relating with five ARGs identified in the MGEs. Similarly, *Acinetobacter* was associated with three ARGs in the MGEs. *Pseudomonas* was correlated with the two ARGs, *bacA* and *mexB*. In contrast, *Thauera* was correlated with *bacA* and *ompR* (Fig. 6C). Our data showed that

some of the ARGs mediated by plasmids and chromosomes were associated with the same taxa (genus level), while ARGs mediated only by chromosomes were found in specific taxa; for example, *abeS* were solely correlated with *Aquimonas* and *mexL* mediated by plasmids were exclusively encoded by the *Rudaea*. Our results generally indicated that many ARGs correlated with the same bacterial genera.

## Uncovering ARG dissemination in potential pathogens

We identified 2,938 VFGs in 26 genera that had at least one ARG. The VFGs were distributed in 253 MAGs. Of these MAGs, 63 were identified as hosts of ARGs, while the remaining MAGs did not contain ARGs (Table S11). To avoid false positives, we filtered out VF genes that were not found in MAG contigs (the cutoff was based on sequence identity ≥70%, *e* value 1e-4, and bit score ≥50). We assigned MAGs as potential pathogens if they contained at least one VFG. More than 89% of MAGs with ARGs showed the potential to be pathogens. These MAGs were affiliated with six phyla (Fig. 7B).

*Klebsiella pneumoniae* and *Enterococcus faecalis* were identified as critical pathogens by the World Health Organization (46). In this study, we identified various ARGs in these species. Of the two species, *Klebsiella pneumoniae* is part of the ESKAPE panel of pathogens (47), carrying six ARGs and multiple ARGs potentially transmitted by MGEs. The most abundant ARG classes carried by potential pathogens were bacitracin, multi-drug, and MLS (Fig. 7A). We also identified the top 11 genera that are potential pathogens based on the number of VFGs encoded in their genomes (Fig. 7A; Table S11), while *Escherichia* coli and *Klebsiella pneumoniae* had a single MAG each that harbored both ARGs and VFGs (Fig. 7A; Table S11). We also identified VFGs in both the plasmids and the chromosomes. Of the 120 VFGs identified in plasmids, 15 potential pathogenic genera were found to host the VFGs. The remaining 2,194 VFGs were found to be carried by chromosomes of 26 genera (Table S12).

## DISCUSSION

Our study on the MAGs provides an overview of the ARGs composition and abundance and their microbial hosts in the AS and WW communities. Furthermore, it provides a risk assessment of potential pathogens in WW treatment plants and highlights the need for continued monitoring to prevent the spread of AMR. We recovered almost 6,000 high- and medium-quality genomes of bacteria and archaea in WW and AS. Recovery of high-quality MAGs allowed a better prediction of functional genes within genomes, including ARGs. However, the number of MAGs recovered from WW metagenomes was 17-fold lower than the MAGs recovered in the AS metagenomes. These differences could be due to several factors, including the limited number of WW metagenomes retrieved from the database, differences in the number of base pairs in various libraries, the disparity in sequencing depth, and differences in library fragment sizes (Table S2). Due to these limitations, our data may not be appropriate to answer questions related to beta diversity. Therefore, we focused on the comparison of the resistome in different taxa. For example, Parks et al. (48) recovered nearly 8,000 MAGs from terrestrial environments without making a beta-diversity inference between the samples. Another study by Feng et al. (49) retrieved MAGs from the chicken gut microbiome from public repositories and predicted ARGs without comparing the diversity of MAGs recovered from each sample and location. In addition, previous studies determined the factors affecting genome recovery in which low sequencing depth was reported as a critical feature in MAGs recovery (18, 50).

Our phylogenetic analysis showed that many MAGs were unclassified, indicating the potential for novel species and genomes in our data set. At higher taxonomic levels, the number of MAGs with known taxonomic classifications increased (e.g., more than 88% of our MAGs were classified at class level). Proteobacteria, Firmicutes, and Bacteroidota were among the top 10 most abundant bacterial phyla in the AS and WW metagenomes. The recovery of genomes from these phyla is interesting because Proteobacteria are

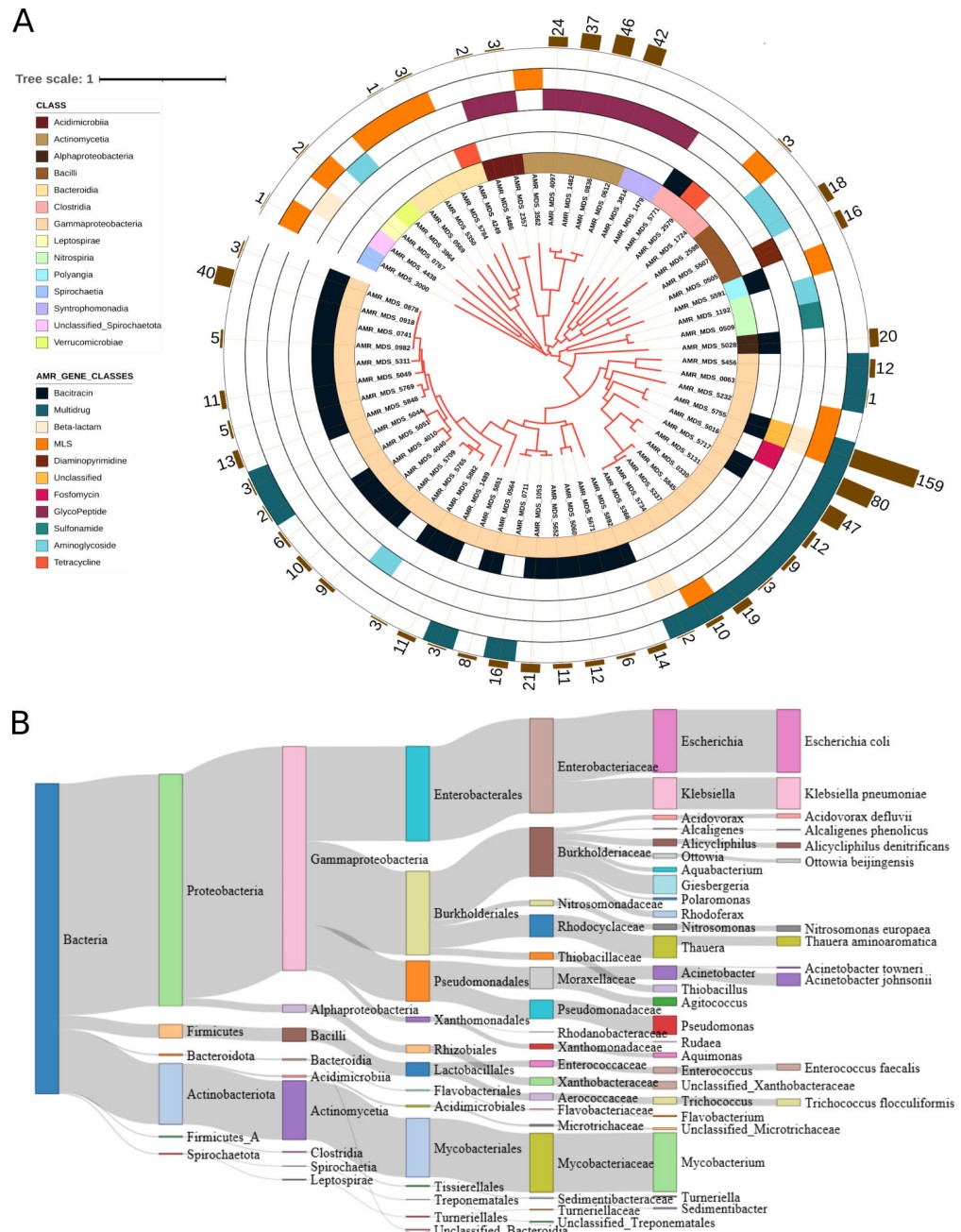

**FIG 7** Phylogenetic tree of the 45 bacterial metagenome-assembled genomes (MAGs) carrying resistomes and host antibiotic-resistant pathogens. (A) Maximum likelihood phylogenetic tree of MAGs constructed using 120 bacterial marker genes in GTDB-Tk. Leaf labels metagenome-assembled genomes sequence. Rings, from the inner to the outside circles, represent Ring 1, which displays the taxonomy of the MAGs at class level, and Rings 2–5, antimicrobial resistance gene classes that are widespread and sparse in bacterial genomes. Bar charts indicate the number of virulence factor genes found in the MAGs with antibiotic resistance genes. The size of the bars shows the number of virulence factor genes detected in MAGs. (B) Range of putative antibiotic resistance pathogens across activated sludge and wastewater samples. The Sankey displays the taxonomy of these putative pathogenic bacteria carrying virulence factor genes. Taxonomy was defined using GTDB-Tk. The size of the bars indicates the relative frequency of potential pathogens.

known to be involved in removing organic pollutants, including nitrogen and aromatic compounds (51). Bacteroidota may participate in acidogenic digestion processes (52), and Firmicutes are syntrophic bacteria capable of degrading various pollutants (53). However, at lower taxonomic rank, the phylogeny of MAGs with known classification

decreased; for example, at the species level, more than 94% of our MAGs were not assigned to any species. This observation is unsurprising as the WW microbial communities are highly diverse, with thousands of unclassified species (10, 54). Haryono et al. (54) recovered MAGs from AS microbial communities and showed that only a small proportion (3.7%) of the MAGs were assigned to the species level, and most of the identified MAGs were classified at higher taxonomic levels than species. Similarly, a study by Singleton et al. (10) used long-read sequencing to recover high-quality MAGs from AS, observing trends similar in phylogenetic classification to our findings, as almost all MAGs were classified at the class level.

In contrast, more than 94% of the MAGs could not be classified at the species level (Table 1). These observations suggest that the microbial communities of WW and AS are still largely unknown. Therefore, similar studies in AS and WW may not only shed light on the structure of the microbial community but also connect genome-centric analysis to the functional potential of these bacteria and archaea communities (10).

Due to our stringent identity cutoffs, our study showed that only 1.06% of the MAGs contained resistomes, while the remaining MAGs did not have resistance genes. This observation is a low proportion considering the number of MAGs (5,916) involved. However, the implications of resistance genes to public health concerns are beyond the quantity of the data, particularly considering these genes are hosted by potential human pathogens and carried by plasmids. Therefore, this paucity of ARGs raises a serious call for urgent research to explore untapped mechanisms of resistance genes at the genomic level in WW treatment plants.

We calculated the ARG distribution in the AS and WW data sets with a genome-centric focus. We observed that AS had more AMR classes than WW, with 22 and 19, respectively. Despite the discrepancies in sample number and sequence depth, we found AMR classes common to AS and WW, including resistance to bacitracin, multiple drugs, MLS, glycopeptide, and aminoglycoside. These antibiotics are among the most common drugs used to treat human pathogens (2) and are found in everyday pharmaceutical products that end up in WW effluents (29, 55). The high abundance of ARGs in AS has been previously reported (29, 55, 56). It is potentially explained by the high concentration of closely knitted microbial communities, favoring AS for disseminating ARGs (10, 29, 56). In addition, current WW treatment plants partially remove ARGs from the effluent (11). Therefore, species that carry ARGs may find their way to the recycled effluent used for agricultural purposes or aquatic environments recharge, where they become a source of ARG dissemination. Our data also showed ARGs conferring resistance to tetracycline, MLS, and sulfonamide in the MAGs, highlighting the importance of monitoring these resistance genes and reducing their co-occurrence in the environment to minimize the risk of ARG spreading within AS and WW microbial communities. Tetracycline, MLS, and sulfonamide resistances are WW effluents' most common ARG classes (29, 57). Nevertheless, comparing ARGs in AS and WW in our study could be biased due to the differences in the number of metagenomes retrieved from the database. More samples are required to fully understand the distribution of ARGs in the AS and WW environments and to determine their impact on human and animal health.

Although further exploration is necessary, our findings suggest that ARG distribution varies among geographical locations. For example, the samples from European countries showed a high diversity of AMR classes, with Germany having a higher abundance of ARG classes than Switzerland and Austria. Meanwhile, Asian countries like Singapore showed a higher abundance of ARG classes compared to China and Japan. The detection of fosfomycin and multi-drug AMR classes from samples in China was interesting because these ARGs are among the ARG classes frequently found in Chinese WW and sewage (11, 55, 56). These results indicate that AMR is a global environmental issue that can only be addressed through an international collective effort (2).

We determined the distribution of ARGs in bacterial communities to investigate the spread of these ARGs in AS and WW with a genome-centric focus. Proteobacteria and Firmicutes were the major hosts of ARGs and carried a multi-AMR class.

Verrucomicrobiota was the only phylum that hosted a single AMR class, while all the other phyla detected hosted at least two AMR classes. As previously reported (4, 11, 58), *Escherichia* sp. was the major pool of ARGs encoding resistance to five AMR classes, including beta-lactam, bacitracin, MLS, and multiple drugs. *Acinetobacter*, *Klebsiella*, *Nitrospira*, *Ottowia*, and *Pseudomonas* genera also host multiple ARG classes. Our results showed that these bacterial genera could be responsible for ARG dissemination within the WW treatment plants and in the environments receiving the effluent (59).

The most prevalent ARG detected in this study was *bacA*, found in more than 13 bacterial genera. The overuse of bacitracin in treating skin infections may have contributed to the high abundance of this gene in our samples. Furthermore, the *bacA* is crucial for the biosynthesis of peptidoglycan and other cell wall components; therefore, bacteria harboring the *bacA* can survive under external stress (29, 60, 61). Our observation aligns with the previous studies of Jia et al. (61), which reported a high abundance of *bacA* in drinking water and WW. In addition, *aph(3′)-I* and *ermF*, conferring resistance to bacitracin and MLS, were hosted by three bacterial genera and were often detected in WWTPs (55).

Interestingly, at lower phylogenic levels (e.g., genus), we observed that some bacterial genera, such as *Acinetobacter*, *Klebsiella*, and *Escherichia*, possess more than one resistance mechanism. This fact makes monitoring ARGs in individual species more difficult. For example, Shi et al. (58) found more than 190 ARGs hosted by *Pseudomonas* with several resistance mechanisms, including multiple resistance mechanisms like efflux pumps, altered target sites, and enzymatic inactivation of antibiotics (62, 63). Additionally, *E. coli* was reported to acquire resistance to multiple antibiotics through HGT and chromosomal mutation, which enabled the species with various resistance mechanisms, including efflux pump, antibiotic target alteration, and enzymatic inactivation of antibiotics (11, 64).

Our analyses show that only 10.26% of the annotated ARGs were located on plasmids, suggesting that most ARGs were not transmitted via plasmids. Some ARGs were found to be associated with plasmids hosted by various bacterial genera, implying the mobility of ARGs between these species. Previous studies identified ARGs mediated by plasmids in various bacterial genera, including *Klebsiella*, *Mycobacterium*, *Escherichia*, and *Enterobacter* (4, 11, 65). These studies identified many ARGs from WW located on plasmids, highlighting the role of MGEs in the spread of ARGs in WW and receiving environments. Zhao et al. (37) showed that ARGs in different MGEs, including plasmids, integrases, and conjugative transposons, are essential in the mobility of ARGs between bacterial communities. The study found that plasmids mediate the transfer of more ARGs than the other MGEs. However, most of the annotated ARGs were located on chromosomes (Fig. 6A), suggesting that these ARGs were transmitted through VGT (inheritance from parent to offspring). A study identified a mobile resistome in WW, where most of the ARGs were associated with chromosomes (35), and only small proportions (10.8%) were found on plasmids, highlighting the involvement of other genetic elements in the spread of ARGs. Shi et al. revealed the potential of VGT through the proliferation of ARG hosts and showed that more ARGs are located on chromosomes than on plasmids (58). Similarly, Dai et al. (11) reported that 22% of the ARGs detected from AS using long reads were located on chromosomes. These observations highlight the potential for VGT of ARGs between bacterial species. While plasmid-borne ARGs are prone to transfer between bacterial species, chromosome-mediated ARGs are the main contributors to the spread of ARGs within bacterial communities.

We explored the health risk of ARGs in WW and AS hosted by human pathogens (66). We identified ARGs in the MAGs containing VFGs. We found that several clinically relevant bacterial genera, including *Escherichia*, *Mycobacterium*, *Klebsiella*, *Enterococcus*, and *Pseudomonas*, carried both VFGs and ARGs. We also found VFGs in potential pathogenic species carrying ARGs located on plasmids, highlighting the threat to human and animal health due to HGT. Chen et al. (57) showed that ARG prediction does not necessarily indicate a potential risk to human health. However, the correlation between

ARGs, MGEs, and human pathogens indicates a severe threat to human health (9, 25, 47). Furthermore, our analyses indicated that many MAGs lacking ARGs harbor VFGs, suggesting they are potential pathogens. Therefore, using our approach could gain insights into the pathogenicity of WW and AS species. These data could help monitor AMR pathogens and develop more effective strategies to control their spread in the environment.

Moreover, previous studies reported that MGEs, such as plasmids and phages, mediated the transfer of VFGs between bacterial species (34). Our results showed that some of the bacterial genera had few VFGs, which indicates that the abundance and types of VFGs in genomes do not determine the pathogenicity of species; in some cases, one VFG might be enough. For example, *Enterococcus faecalis* is a critical pathogenic bacteria causing several nosocomial infections (67, 68). Our results showed that this species had few VFGs, which indicates that some species can be pathogens even if the number of VFGs in their genome is low. The ESKAPE panel refers to six bacterial species considered the most significant contributors to antimicrobial resistance and nosocomial infections (47). In our data set, these bacteria encoded multiple resistance mechanisms, upsurging their pathogenic potential and health risk. Therefore, it is crucial to monitor the risk of ARGs by identifying the VFGs to predict the potential pathogenicity and to prevent their spread in healthcare settings and environments.

## Conclusions

Our study analyzed MAGs recovered from AS and WW treatment plants. The taxonomic composition of the MAGs was determined, and the results showed that the microbial communities and ARG distribution varied in the AS and WW. Our data also showed that ARGs were widely distributed, and the presence of ARGs in various geographical locations poses a significant public health concern. We also revealed that the most dominant AMR classes in the MAGs were bacitracin, multi-drug, MLS, and aminoglycosides. The results indicated that bacterial communities and MGEs are crucial in disseminating ARGs and that there is potential HGT of ARGs through plasmids hosted by different bacterial genera. Our study showed the presence of ARGs and VFGs in the MAGs, which could have dire implications for human and animal health and cause contagious diseases that could be difficult to treat and control. Further studies with larger data sets encompassing more geographical locations collected in a time series are needed to monitor changes in the long-term prevalence of ARGs and to design new strategies for reducing the dissemination of ARGs in natural and constructed ecosystems.

## MATERIALS AND METHODS

### Selection of metagenomes from activated sludge and wastewater samples

A total of 165 publicly available metagenomes were collected from the SRA derived from AS and WW treatment plants from the Terrestrial Metagenome Database biome ("activated sludge" and "wastewater" or "wastewater treatment plants") (19). This data set is a part of the Collaborative Multi-domain Exploration of Terrestrial metagenomes (CLUE-TERRA) consortium (https://www.ufz.de/index.php?en=47300). Metagenomics samples in the CLUE-TERRA consortium have previously been filtered based on the following criteria: (i) because non-metagenomics libraries in the SRA can be wrongfully annotated as metagenomics, only true whole-genome shotgun libraries were kept, which was achieved using PARTIE (69), using default parameters; (ii) metagenomes with sequence quality scores below 60%, determined using SRA-Tinder (https://github.com/NCBI-Hackathons/SRA_Tinder) with default parameters, were discarded; (iii) to allow for comparative studies, only metagenomes sequenced using the Illumina sequencing platform and with a minimum of 8 million paired-end reads per library were kept; and (iv) given the CLUE-TERRA consortium's focus on terrestrial environments, all libraries containing coordinates or terms for sea environments were excluded. Metadata

information from each metagenome was collected from a standardized metadata database (19) and a set of data fields at the initial collection. The metadata were later manually curated to get the other missing information. This includes the exact location of samples and information about the different WW and AS. Overall, the data set covers 13 different locations. All these and other detailed information about the data set are provided in Table S1.

## Metagenome quality control and assembly

The raw sequence data were quality-checked and assembled using metaSPades (70). MAGs recovery was performed with metaWrap v.0.7 (71), and phylogenetic analyses were done using GTDB-Tk (44). The MAGs were dereplicated to genome operational taxonomic units using MuDoGer (20). A detailed description of these steps can be found in the supplemental methods and Fig. S7.

## Prediction of ARGs and MGEs

We generated ARGs profiles of the recovered MAGs using deepARG v.1.0.2 (72). We selected the deepARG v.2.0 (72) tool because it is a deep learning model designed to specifically predict ARGs belonging to over 30 ARG classes, and the database was manually curated resources of ARGs. The sequences from the genomes were translated from nucleotide to amino acid using the faTrans tool from KentUlis (73). The amino acid sequences of the genomes were then used as input files for alignment against the DeepARG–DB (72) database using blastp implemented in the DIAMOND software v.0.9.17.118 (74). The sequences of each MAG were as defined as ARG-like open reading frames (ORFs) at the $e$ value cutoff threshold of 1e-10, probability of point of 0.8, and ≥80% sequence identity to reduce the risk of false-positive ARG-like sequences. A cutoff of 80% identity allows for predicting high-quality and strictly identifying ARGs (72). However, because we are working with MAGs that largely reflect the uncultured and thus less known proportion of AS and WW prokaryotes, it is fitting to allow for the prediction of novel ARGs within our data. We converted all ARG identifiers to upper case to standardize the IDs and remove the differentiation between gene and protein identifiers. Plasflow (75) was further used to predict the occurrence of plasmids in the MAGs, emphasizing whether the ARGs are located on plasmids or chromosomes. The resistance mechanisms of the ARGs were identified from the CARD (76).

## Co-occurrence of ARGs and VFGs

To determine whether ARG predicted in MAGs and MGEs are hosted by potentially pathogenic bacteria, we mapped the sequences to the virulence factors database of Victors (a manually curated database for VFGs of human and animal pathogens) (40) through a blastp implement in diamond v.0.9.22 (74). The VFGs present in the MAGs and MGEs were filtered to overcome false-positive results of pathogenic bacteria where they are not. Herein, three criteria were adopted: sequence identity ≥70%, $e$ value of 1e-4, and bit score of ≥50. The results of the number of potentially pathogenic bacteria carrying resistance genes were displayed as bar plots in the phylogenetic tree.

## Statistical analysis

The prevalence of ARGs and AMR classes in MAGs was calculated based on the presence and absence of each ARG in each MAG at different taxonomic levels. The prevalence of each MAG was calculated as the number of taxa containing the ARGs or AMR class divided by the total number of MAGs belonging to the respective taxa, similar to the prevalence definition by Danko et al. (33). Student's $t$-test was employed to identify the significant difference between the ARGs predicted in AS and WW samples. The correlation between the number of ARGs in an ARG class and the number of MAGs with at least one ARG from an AMR class was calculated using Kendall's rank correlation.

Furthermore, network analysis was used to explore the detailed relationships between microbial communities and ARGs found in MGEs based on the correlation matrices constructed by calculating all possible pairwise Spearman correlation coefficients between ARGs and the bacterial genera. All analyses were performed using R v.3.5.2 packages, such as ggplot2, pheatmap, and Tidyverse. Gephi v.0.9.2 was employed for network visualization (77).

## ACKNOWLEDGMENTS

We thank the administration and support staff of EVE which kept the system running and supported us with our scientific computing needs: Thomas Schnicke, Ben Langenberg, Guido Schramm, Toni Harzendorf, Tom Strempel, and Lisa Schurack from the UFZ, and Christian Krause from iDiv. We also thank Rodolfo Brizola Toscan for the computational support during the metagenomics data analysis.

This study is supported by the Helmholtz Young Investigator grant VH-NG-1248 Micro "Big Data" and the Deutsche Forschungsgemeinschaft (German Research Foundation) project number 460129525. N.A. was supported with a scholarship funded by Petroleum Technology Development Fund and German Academic Exchange Service number 91717355.

U.R. conceptualized the idea; N.A. performed data generation and processing, performed computational analyses, interpreted the results, and drafted the original article; J.P.S., J.Z., H.H., and U.R. supervised the work for formal data analysis; S.S., H.H., J.Z., O.G., J.P.S., and U.R. critically revised the manuscript. All authors reviewed and agreed on the final version of the manuscript.

## AUTHOR AFFILIATIONS

[1]Department of Environmental Microbiology, Helmholtz Center for Environmental Research-UFZ, Leipzig, Germany
[2]Department of Biochemistry, Faculty of Natural Science, University of Leipzig, Leipzig, Germany
[3]Department of Isotope Biogeochemistry, Helmholtz Centre for Environmental Research-UFZ, Leipzig, Germany
[4]Research Center for Eco-Environmental Sciences, Chinese Academy of Sciences, Beijing, China
[5]Institute of Water Chemistry, Technical University of Dresden, Dresden, Germany
[6]Zuckerberg Institute for Water Research, J. Blaustein Institutes for Desert Research, Ben Gurion University, Midreshet Ben Gurion, Israel

## AUTHOR ORCIDs

Nafi'u Abdulkadir http://orcid.org/0000-0003-1264-2003
Hauke Harms http://orcid.org/0000-0002-7316-7341
Ulisses Rocha http://orcid.org/0000-0001-6972-6692

## FUNDING

| Funder | Grant(s) | Author(s) |
| --- | --- | --- |
| Helmholtz Association (亥姆霍兹联合会致力) | VH-NG-1248 Micro' Big Data' | Ulisses Rocha |
| Deutsche Forschungsgemeinschaft (DFG) | 460129525 | Ulisses Rocha |
| German Academic Exchange Service (DAAD) | 91717355 | Nafi'u Abdulkadir |

## AUTHOR CONTRIBUTIONS

Nafi'u Abdulkadir, Data curation, Formal analysis, Investigation, Methodology, Valida-tion, Visualization, Writing – original draft, Writing – review and editing | Joao Pedro Saraiva, Formal analysis, Methodology, Supervision, Writing – original draft | Stefan Stolte, Conceptualization, Writing – original draft, Writing – review and editing | Ulisses Rocha, Conceptualization, Formal analysis, Funding acquisition, Investigation, Methodol-ogy, Project administration, Resources, Software, Supervision, Validation, Visualization, Writing – original draft, Writing – review and editing.

## DATA AVAILABILITY

The metagenome-assembled genomes (MAGs) recovered in this study are deposited at NCBI under BioProject no. PRJNA925477. The individual links for each MAG accession number may be found in Table S13. The complete set of ARGs (and their sequences in fasta format) are available in the long-term data archive at the Helmholtz Center for Environmen-tal Research-UFZ data center (https://www.ufz.de/record/dmp/archive/14020/de/).

## ADDITIONAL FILES

The following material is available online.

### Supplemental Material

**Supplemental figures (Spectrum02918-23-S0001.pdf).** Fig. S1 to Fig. S7.
**Supplemental text (Spectrum02918-23-S0002.pdf).** Supplemental methods.
**Supplemental tables (Spectrum02918-23-S0003.xlsx).** Tables S1 and S3 to S12.
**Table S2 (Spectrum02918-23-S0004.pdf).** Summary of sequence read characteristics.
**Table S13 (Spectrum02918-23-S0005.xlsx).** Metagenome-assembled genome (MAG) accession numbers provided by the NCBI.

### Open Peer Review

**PEER REVIEW HISTORY (review-history.pdf).** An accounting of the reviewer comments and feedback.

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
