## [Reviewer comments · Microbiology Spectrum]

Microbiology Spectrum

Genome-centric analyses of 165 metagenomes show that mobile genetic elements are crucial for the transmission of antimicrobial resistance genes to pathogens in activated sludge and wastewater

Nafi'u Abdulkadir, Joao Pedro Saraiva, Junya Zhang, Stefan Stolte, Osnat Gillor, Hauke Harms, and Ulisses Nunes da Rocha

Corresponding Author(s): Ulisses Nunes da Rocha, Helmholtz-Zentrum fur Umweltforschung UFZ

Review Timeline:

Submission Date:	July 28, 2023
Editorial Decision:	August 24, 2023
Revision Received:	September 30, 2023
Accepted:	November 25, 2023

Editor: Adriana Rosato

Reviewer(s): The reviewers have opted to remain anonymous.

Transaction Report:

DOI: <https://doi.org/10.1128/spectrum.02918-23>

August 24, 2023

Dr. Ulisses Nunes da Rocha
Helmholtz-Zentrum für Umweltforschung UFZ
Permoserstraße 15
Leipzig, N/A 4318
Germany

Re: Spectrum02918-23 (Genome-centric analyses of 165 metagenomes show that mobile genetic elements are crucial for the transmission of antimicrobial resistance genes to pathogens in activated sludge and wastewater)

Dear Dr. Ulisses Nunes da Rocha:

Your manuscript has been reviewed by an expert in the field and myself; we both found the study to be interesting. However, there are a couple of unclear points that require attention.

Link Not Available

Sincerely,

Adriana Rosato

Journals Department
Reviewer comments:

Reviewer #1 (Comments for the Author):

This study determined the presence of ARGs in metagenome-assembled genomes (MAGs) recovered from AS and WW treatment plants and evaluated the risks of MAGs-carrying ARGs in potential human pathogens. This study also analyzed whether ARGs and virulence factor genes (VFGs) in MAG are present on chromosomes or plasmids. This manuscript is original and generally well written. This article has advanced experimental methods and rigorous writing. The main issue is the "INTRODUCTION", which requires a detailed review of previous research progress and existing shortcomings.

Other comments are as follows:

- 1.Line 2 and 58: please make sure to use "antimicrobial resistance genes" instead of "antibiotic resistance genes".
- 2.Line 24: contamination <10%.
- 3.Line 40: the names of genes need to be uniformly italicized in context, such as aph(3')-I. Please carefully check for other genes.
- 4.Line 82: such research methods have also been done in previous studies , for example in the field of fermented foods (such as sufu) in 2021. It is appropriate to supplement this literature here.
- 5.Line 108: 14 locations? The figure 1 shows 13 positions.
- 6.Line 122: 2660 should be expressed as 2,660. Pay attention to other numerical expressions in manuscript.
- 7.Line 129: should be Fig. S1 and Table S4.
- 8.Line 132: change GTDB-tk to GTDB-Tk.
- 9.Line 141: why is the number of MAGs at the genus level in Table 2 not 5916?
- 10.Line 170: change Figure 1 to Fig. 1.
- 11.Line 197: change Fig. 3b to Fig. 3B. The lowercase letters after the numerical value need to be changed to uppercase letters. Please note that the entire text should be expressed in this way.
- 12.Line 296: 94%?
- 13.Line 450: there are many mobile genetic elements (MGEs), such as plasmids, bacteriophages, transposons, and insertion sequences. Why did this article only consider plasmids?
- 14.Line 549-550: pay attention to uppercase or lowercase letters of the titles in the references. Many similar errors occur in the references. Please check carefully in context.
- 15.Line 810: 14 different locations?
- 16.Line 846: plasmids and chromosomes.
- 17.Line 860: activated sludge or wastewater.
- 18.Line 864: change GTDB-tk to GTDB-Tk.

Staff Comments:

Preparing Revision Guidelines

Please return the manuscript within 60 days; if you cannot complete the modification within this time period, please contact me. If you do not wish to modify the manuscript and prefer to submit it to another journal, please notify me of your decision immediately so that the manuscript may be formally withdrawn from consideration by Microbiology Spectrum.

Manuscript Spectrum02918-23

Dear Dr. Adriana Rosato,

We appreciate the reviewers' comments and yours for the essential remarks and suggestions to improve our manuscript.

We have considered all the comments in the revised version of the manuscript and indicated the changes in the point-by-point reply. In short, the introduction to the manuscript has been updated with comprehensive literature on antimicrobial studies. We have also identified the shortcomings and suggested future research in the field. We also supplemented the introduction with studies using similar approaches to ours in other environmental compartments, such as fermented food, which were added to indicate the advancement of genome-centric analysis in antibiotic resistance genes studies.

We hope the point-by-point replies are clear and address all reviewer's comments, and we are looking forward to your reply.

Sincerely,

Ulisses Nunes da Rocha

Point-by-point reply to the reviewers' comments:

Reply to comments by Reviewer #1:

Reviewer #1 (Comments for the Author):

Intro: This study determined the presence of ARGs in metagenome-assembled genomes (MAGs) recovered from AS and WW treatment plants and evaluated the risks of MAGs-carrying ARGs in potential human pathogens. This study also analyzed whether ARGs and virulence factor genes (VFGs) in MAG are present on chromosomes or plasmids. This manuscript is original and generally well written. This article has advanced experimental methods and rigorous writing.

General comments by the reviewer:

The main issue is the “**INTRODUCTION**”, which requires a detailed review of previous research progress and existing shortcomings.

Reply to general comments:

We thank the reviewer for the kind comments and understanding of the issues in the introduction, which lacks a detailed review of the previous research and challenges in antimicrobial research. We added a detailed review of the literature and identified the challenges in the AMR research.

The previous research progress added in the introduction of the revised version of the manuscript is as follows:

1. WWTPs are considered an environmental hotspot for disseminating ARGs due to the presence of mobile genetic elements and diverse ARBs. The high abundance of bacterial communities in the WW facilitates the direct route for the distribution of ARGs in the environment. This addition of previous research progress appeared in the revised version of the manuscript in lines 66 – 67.
2. Previous studies showed that WWTPs are the most important sources of resistance genes due to the secretion of antibiotic residues from human waste, veterinary and hospitals. Therefore, surveillance of ARGs is necessary in WWTP as part of the effort to diminish the emergence and distribution of resistance in the ecological environment and the possibility of detecting new bacterial resistance mechanisms. It is also essential to understand the various mechanisms by which bacterial species develop resistance to antibiotics for the establishment of policies

to fight resistance. This addition of previous research progress appeared in the revised version of the manuscript in lines 71 – 76.

3. Additionally, previous studies reported the occurrence of ARGs in the treated WW and their distribution into the receiving environment, which further revealed that continuous discharge of poorly treated WW could enable the transfer of resistance genes to pathogenic bacteria and spread ARBs in the environments. The existence of ARGs and ARBs in the activated sludge, effluents and influents of WWTPs from various countries shows the global distribution of resistance genes in the environment. This addition of previous research progress appeared in the revised version of the manuscript in lines 77 – 81.
4. Recently, Nava and co-workers revealed that multi-resistance bacteria are present in WW and distributed into the environment through effluent discharge, which may lead to the development of “Superbug” species. The existence of multi-resistance species in WW is supported by the co-association of antibiotics, heavy metals, ARBs, and ARGs in WWTPs. Therefore, an urgent need to monitor heavy metal resistance in the WWTP and design adequate strategies for assessing the risks of ARGs and heavy metal resistance in ecological settings is necessary. This addition of previous research progress appeared in the revised version of the manuscript in lines 82 – 87.
5. Studies on metagenomics analysis unveiled the presence of important clinical ARG classes in activated sludge, including penicillin, tetracycline, sulfonamides and others that remain in treated wastewater. A study by Cacace showed the abundance of ARGs in treated WW and receiving bodies from WWTP effluent in ten euro European countries. The study showed the presence of ARGs in all effluents and river water samples, demonstrating a complex method of acquiring ARGs in different bacterial communities. Recently, Talat and co-workers provided a comprehensive overview of ARGs in hospital wastewater using metagenomics. The study uncovered many important clinical resistance gene classes, including beta-lactam, aminoglycosides, macrolide carbapenem and sulfonamides, which were hosted by human pathogens, such as *Pseudomonas aeruginosa*, *Acinetobacter baumannii* and *Klebsiella pneumoniae*, evinced the peril associated with ARGs transmission in the environments. This addition of previous research progress appeared in the revised version of the manuscript in lines 106 – 116.

6. Recently, genome-centric approaches have been used to determine the distribution patterns of resistance genes in urban, fermented foods and sewage to manage ARGs as a public health problem. A previous study by Tan and colleagues identified the distribution patterns of ARGs in sulfur bacterial communities. Further, it revealed that ARGs were enriched in opportunistic pathogens. The study revealed the presence of the most critical clinical drug classes used to treat diseases. It showed that MAGs uncovered significant pathways of human resistance gene consumption due to the uptake of ready-to-eat food. The study by Junya Zhang revealed the profile of ARGs in combined sewage overflows and recovered MAGs conferring resistance to many ARGs, including *bacA*, *acrA*, *rsmA* and *mexK OmpA*, among others. This addition of previous research progress appeared in the revised version of the manuscript in lines 128 – 136.

The existing shortcomings that were added to the revised version of the manuscript are:

- a) One of the challenges of AMR research is the lack of a standard method for quantification and surveillance of ARG acquisition, despite the richness of ARGs in the WW effluents and transmission of genes from pathogens to commensal species in the environment facilitated by mobile genetic elements (MGEs) MGE via horizontal gene transfer (HGT) or vertical gene transfer (VGT). This addition of the existing shortcomings in antimicrobial research appeared in the revised version of the manuscript in lines 88 – 91.
- b) The studies of Nava et al., and Larsson and Flach, further revealed no standard method for removing ARG in the environment, including WWTPs. Therefore, strategies involving biotic and microbial remediation are needed to mitigate the evolutionary selection of ARGs. To tackle the menace of ARGs in the environment, public awareness about the reasonable usage and pernicious upshot of antibiotic misuse and abuse should be implemented to reduce antibiotic dissipation. This addition of the existing shortcomings in antimicrobial research appeared in the revised version of the manuscript in lines 91 – 96.

Other specific comments are as follows:

Comment 1:

Line 2 and 58: please make sure to use “antimicrobial resistance genes” instead of “antibiotic resistance genes”.

Reply 1: We thank the reviewer for the comment. We use the term “antimicrobial resistance genes as suggested. The revised version of the manuscript is indicated changes in lines 2 and 58.

Comment 2:

Line 24: contamination <10%.

Reply 2: We thank the reviewer for the comment. We use the less than sign (<10%) to indicate the level of contamination in the recovered metagenome-assembled genomes (MAGs). In the updated version of the manuscript, the changes were reflected in line 24.

Comment 3:

Line 40: the names of genes need to be uniformly italicized in context, such as aph(3')-I. Please carefully check for other genes.

Reply 3: We thank the reviewer for the comment. We checked and italicized the names of resistance genes in places where they were not italicized before throughout the manuscript. In the revised version of the manuscript, such changes can be seen in lines 40, 240-241 and 256.

Comment 4:

Line 82: such research methods have also been done in previous studies, for example in the field of fermented foods (such as sufu) in 2021. It is appropriate to supplement this literature here.

Reply 4: Thank you for the information regarding the relevance of supplementing our manuscript with previous studies conducted in fermented foods, such as sufu, using genome-centric analysis. We appreciate your input, and we addressed this comment. We agree that supplementing relevant studies in the introduction can provide a broader context for our research. This literature would help establish a more comprehensive understanding of the complexity of antimicrobial research and show the continuity of antimicrobial studies. We carefully read the manuscript and supplement the literature in the introduction in lines 128 -134.

Comment 5:

Line 108: 14 locations? The figure 1 shows 13 positions.

Reply 5: Thank you for your attention to detail and comment regarding Figure 1. We appreciate your thorough review of our work. You are correct; Figure 1 indeed depicts 13 positions, not 14. We apologize for any confusion in the figure's description and appreciate your diligence in pointing out this discrepancy. We have also thoroughly reviewed the data and ensure the accuracy of this

information. We have shown these changes in the revised manuscript in lines 154, 215, 476 and 878.

Comment 6:

Line 122: 2660 should be expressed as 2,660. Pay attention to other numerical expressions in a manuscript.

Reply 6: We thank the reviewer for the valuable comment. We have carefully checked the manuscript and updated the text with the standard numerical expressions as suggested by the reviewer. These changes appeared in the revised version of the manuscript in lines 165, 168, 172, 178 and 349.

Comment 7:

Line 129: should be Fig. S1 and Table S4.

Reply 7: We thank the reviewer for suggesting improving our manuscript. We have changed the initial presentation of Fig. S1; Table S4 to Fig. S1 and Table S4 as suggested by the reviewer. We showed the changes in line 175 of the revised version of the manuscript.

Comment 8:

Line 132: change GTDB-tk to GTDB-Tk.

Reply 8: We have changed the lowercase letter “t” of GTDB-tk to the uppercase letter “T” GTDB-Tk, as suggested by the reviewer. This change was reflected in the revised version of the manuscript in lines 178, 480, 519, 887, 918, 925, and 932.

Comment 9:

Line 141: why is the number of MAGs at the genus level in Table 2 not 5,916?

Reply 9: Thank you for your insightful feedback regarding Table 1 and the number of MAGs at the genus level. We appreciate your diligence in reviewing our manuscript. After carefully reviewing your comment and re-checking the data, we have identified an error in the previous version of Table 2, which led to an incorrect number of MAGs with unknown taxonomic

classification at the genus level. We apologize for any confusion this may have caused. We have now rectified this issue, and the revised Table 2 accurately reflects 4,182 as the number of MAGs with unknown classification at the genus level. As expected, the total number of MAGs with known and unknown taxonomic classification are 5,916 at the genus. We sincerely appreciate your valuable input, as it has significantly improved the quality and accuracy of our manuscript.

Comment 10:

Line 170: change Figure 1 to Fig. 1.

Reply 10: We have changed Figure 1 to Fig. 1. In the revised version of the manuscript, such changes can be seen in line 216.

Comment 11:

Line 197: change Fig. 3b to Fig. 3B. The lowercase letters after the numerical value need to be changed to uppercase letters. Please note that the entire text should be expressed in this way.

Reply 11: We have changed the lowercase letters after the numerical value to uppercase in all the places it appeared in the manuscript. In the revised version of the manuscript, such changes can be seen in lines 227, 238, 243, 246, 249, 263, 265, 269, 274, 297, 302-305 and 413.

Comment 12:

Line 296: 94%?

Reply 12: We have carefully checked the data and noticed that the percentage of MAGs with unknown phylogeny is 94%, not 92%. The text has been corrected in the revised version of the manuscript in lines 334 and 342.

Comment 13:

Line 450: there are many mobile genetic elements (MGEs), such as plasmids, bacteriophages, transposons, and insertion sequences. Why did this article only consider plasmids?

Reply 13: Thank you for your thoughtful question regarding our study's focus on plasmids while excluding other mobile genetic elements. We appreciate your inquiry and would like to provide a clear rationale for this decision.

Our study was designed with a specific research focus on understanding the role of plasmids in horizontal gene transfer within a wastewater bacterial community. By narrowing our goal to plasmids, we maintained a straightforward research question and hypothesis, facilitating a more comprehensive investigation. Plasmids are known to be significant vectors for gene transfer in many bacterial populations than any other mobile genetic elements (<https://doi.org/10.1016/j.scitotenv.2021.150095>). They mostly carry genes that provide selective advantages, such as antibiotic resistance or metabolic capabilities. Our research aimed to address questions related to the transmission of specific antibiotic resistance genes within a bacterial community, and plasmids were highly relevant to this investigation. However, we acknowledge that other mobile genetic elements, such as transposons, insertion sequences and phages, also play essential roles in horizontal gene transfer and bacterial evolution. In the discussion section of our paper, we showed that other genomics content, such as chromosomes, also played a role in the transfer of resistance genes via vertical transfer. We have added the reason for concentrating only on plasmids in the revised version of the manuscript. The changes can be seen in lines 122-123.

Comment 14:

Line 549-550: pay attention to uppercase or lowercase letters of the titles in the references. Many similar errors occur in the references. Please check carefully in context.

Reply 14: Thank you for the valuable comment. We have checked the entire references and updated the titles with lowercase letters. In the revised version of the manuscript, such changes can be seen in lines 596-597, 625-626, 628-629, 633, 647-648, 659, 668-670, 757-758, 765-767, 774-775, 813-816, 832-833 and 845-846.

Comment 15:

Line 810: 14 different locations?

Reply 15: Thank you for your attention to detail and your comment regarding the number of locations. We appreciate your thorough review of our work. You are correct; the number of

locations is 13, not 14. We apologize for any confusion and appreciate your diligence in pointing out this discrepancy. Your feedback is valuable to us, and we are grateful for your contribution to ensuring the accuracy of our research. We have also thoroughly reviewed the data and methodology to ensure the accuracy of this information. In the revised version of the manuscript, such changes can be seen in lines 154, 215, 476 and 878.

Comment 16:

Line 846: plasmids and chromosomes.

Reply 16: We have changed the uppercase letter of plasmid and chromosomes to lowercase letters. The changes can be seen in the revised version of manuscript line 914.

Comment 17:

Line 860: activated sludge or wastewater.

Reply 17: We have changed the uppercase letter of activated sludge or wastewater to lowercase letters. The changes can be seen in the revised version of manuscript lines 928-929.

Comment 18:

Line 864: change GTDB-tk to GTDB-Tk.

Reply 18: We have changed the lowercase letter "t" of GTDB-tk to the uppercase letter "T" GTDB-Tk, as suggested by the reviewer. This change was reflected in the revised version of the manuscript in lines 178, 480, 519, 887, 918, 925, and 932.

Re: Spectrum02918-23R1 (Genome-centric analyses of 165 metagenomes show that mobile genetic elements are crucial for the transmission of antimicrobial resistance genes to pathogens in activated sludge and wastewater)

Dear Dr. Ulisses Nunes da Rocha:

Your manuscript has been accepted, and I am forwarding it to the ASM production staff for publication. Your paper will first be checked to make sure all elements meet the technical requirements. ASM staff will contact you if anything needs to be revised before copyediting and production can begin. Otherwise, you will be notified when your proofs are ready to be viewed.

Sincerely,
Adriana Rosato
Editor
Microbiology Spectrum

Reviewer #1 (Comments for the Author):

This article has been revised according to the reviewer's suggestions and is suitable for publication.